# Fake it to make it: Using synthetic data to remedy the data shortage in joint multimodal speech-and-gesture synthesis

## Abstract

*Although humans engaged in face-to-face conversation simultaneously communicate both verbally and non-verbally, methods for joint and unified synthesis of speech audio and co-speech 3D gesture motion from text are a new and emerging field. These technologies hold great promise for more human-like, efficient, expressive, and robust synthetic communication, but are currently held back by the lack of suitably large datasets, as existing methods are trained on parallel data from all constituent modalities. Inspired by student-teacher methods, we propose a straightforward solution to the data shortage, by simply synthesising additional training material. Specifically, we use uni-modal synthesis models trained on large datasets to create multimodal (but synthetic) parallel training data, and then pre-train a joint synthesis model on that material. In addition, we propose a new synthesis architecture that adds better and more controllable prosody modelling to the state-of-the-art method in the field. Our results confirm that pre-training on large amounts of synthetic data improves the quality of both the speech and the motion synthesised by the multimodal model, with the proposed architecture yielding further benefits when pre-trained on the synthetic data.*

## 1. Introduction

Human beings are embodied, and we use a wide gamut of the expressions afforded by our bodies to communicate. In concert with the lexical and non-lexical (prosodic) components of speech, humans also leverage gestures realised by face, head, arm, finger, and body motion – all driven by a shared, underlying communicative intent [58] – to improve face-to-face communication [30, 66].

Research into automatically recreating different kinds of human communicative behaviour, whether it be speech audio from text [85], or gesture motion from speech [92], have a long history, as these are key enabling technologies for, e.g., virtual agents, game characters, and social robots

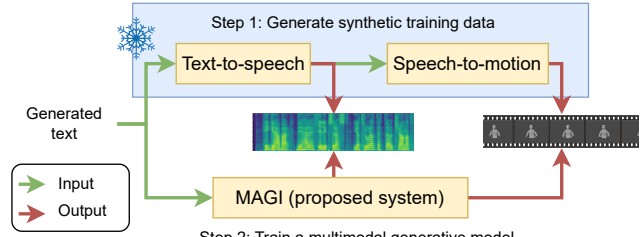

Figure 1. MAGI: Multimodal Audio and Gesture, Integrated

[14, 41, 57, 68]. The advent of deep learning has led to an explosion of research in the two fields [54, 66, 83]. Gesture synthesis, in particular, has been shown to benefit from access to both lexical and acoustic representations of speech [3, 42, 43, 104]. That said, joint and simultaneous synthesis of both speech and gesture communication (pioneered in [78]) remains severely under-explored. This despite the fact that simultaneously generating both modalities together not only better emulates how humans produce communicative expressions, but also offers a stepping stone towards creating non-redundant gestures that can complement and even replace speech, like human gestures do [34]. On top of this, recent research efforts towards integrating the synthesis of the two modalities have demonstrated improvements in coherent [6, 62], compact [62, 94], jointly and rapidly learnable [61], convincing [61, 62], and cross-modally appropriate [62] synthesis of speech and 3D gestures from text.

The current state of the art in joint multimodal speech-and-gesture synthesis, Match-TTSG [62], achieves strong performance via modern techniques such as conditional flow matching (OT-CFM) [51] with U-Net Transformer [91] encoders [77]. However, there still remains a noticeable gap between synthesised model output and recordings of natural human speech and gesticulation [62]. This contrasts with recent breakthroughs in "generative AI", which can synthesise text [2, 13], images [77], and speech audio [80, 84] that all are nigh indistinguishable from those created by humans. The critical difference is that whereas those strong models for synthesising single modalities benefit from training on vast amounts of data (cf. [27]), exist-

ing parallel datasets of speech audio, text transcriptions, and human motion are radically smaller. This is especially true if we require good motion quality (which at present generally necessitates high-end 3D motion capture) and speech audio with a spontaneous character and quality suitable for speech synthesis. The state-of-the-art joint synthesis system demonstrated in [62] was thus trained on 4.5 hours of parallel speech and gesture data from [22]; larger parallel corpora exist [49, 53], but exhibit some quality issues (cf. [44]) and do not exceed 100 hours, a far cry from the corpora used to train leading generative AI systems. It stands to reason that multimodal synthesis systems could gain substantially from overcoming the limitations imposed by training only on presently available parallel corpora.

In this paper, we propose two improvements to the state-of-the art multimodal speech-and-gesture synthesis:

1. We pre-train a joint speech-and-gesture synthesis model on a large parallel corpus of *synthetic* training data created using leading text, text-to-speech, and speech-to-gesture systems (Fig. 1). This provides a straightforward way to let multimodal models benefit from advances in data and systems for unimodal synthesis.

2. We extend [62] with a probabilistic duration model (similar to [48]) and individual models of pitch and energy (similar to [75]). This enables more lifelike and more controllable synthetic expression.

The resulting joint synthesis system is orders of magnitude smaller and faster than the models used for synthesising the pre-training data. Our subjective evaluations show that the proposed pre-training on synthetic data improves the speech as well as the gestures created by a joint synthesis system, and that the architectural modifications further benefit a system pre-trained on large synthetic data and also enable output control. For examples of model output, please see our anonymous webpage at cvprhumogen24.github.io/MAGI/; code will be released with future versions of the paper.

## 2. Background

In this section, we review synthesis of text, speech audio, and 3D gesture motion, along with existing work in multimodal speech-and-gesture synthesis. For each task, we state how the methods relate to our contributions and briefly discuss how synthetic data can improve synthesis models.

### 2.1. Text generation

The rise of large language models (LLMs) has brought revolutionary improvements to text generation. Transformer-based [91] LLMs using Generative Pretrained Transformers (GPTs) [71] like [2, 13, 88] are capable of generating text virtually indistinguishable from that written by humans.

The critical methodological advances for LLMs are pre-training on vast amounts of diverse data, coupled with fine-tuning on a small amount of high-quality, in-domain material, e.g., via Reinforcement Learning from Human Feedback (RLHF) [9]. This methodology of pre-training foundation models followed by fine-tuning on the best data has been validated to give excellent results across several modalities [11, 111]. In this paper, we for the first time use that methodology in joint speech-and-gesture synthesis.

Fine-tuned LLMs allow generating of diverse text samples for many domains through *prompting* the model, i.e., providing a written text prompt at runtime describing the output to generate. Prompting has been useful for many tasks including creating synthetic dialogue datasets [1] and selecting appropriate gestures based on verbal utterances [28]. We use this ability to create an arbitrarily large material of conversational text sentences in the style of a given speaker/corpus as a basis for our synthetic-data creation.

### 2.2. Speech synthesis

Recent advancements in deep generative modelling have significantly improved text-to-speech (TTS) [83], achieving levels of naturalness that rival recorded human speech [80, 84]. TTS approaches are primarily divided into two broad classes: autoregressive (AR) and non-autoregressive (NAR) architectures. AR architectures produce acoustic outputs sequentially, using mechanisms such as neural cross-attention [10, 15, 50, 79, 110] or neural transducers [59, 60, 101] to connect inputs symbols to the outputs. Conversely, non-autoregressive models [25, 36, 37, 48, 63, 69, 75, 112] generate the entire utterance in parallel. The NAR approach is typically faster, especially on GPUs, but AR methods (which invest more computation into synthesis) often have the edge in synthesis quality.

Recently, there has been a trend [10, 12, 15, 46, 93] to quantise audio waveforms into discrete tokens [16, 46], and then adapt an LLM-like autoregressive approach (e.g., with GPTs) to learn to model these audio tokens on large datasets. Synthesised token sequences can subsequently be converted back to audio [81]. Speaker and style adaptation can be achieved by seeding (prompting) the model with an audio snippet, something we leverage to create diverse stochastic synthetic training data for our work.

LLM-like TTS can give exceptional results when trained on large datasets, but models risk confabulating (similar to well-known issues with LLMs) and getting trapped in feedback loops due to the autoregression [10, 15]. Our paper therefore describes a pipeline for mitigating these problems when creating synthetic training data at scale.

In NAR TTS, it has been found that conditioning the TTS on the output of a model of prosodic properties, e.g., per-phone pitch and energy, can benefit synthesis [67, 75, 112]. This furthermore affords control over speech output by replacing or manipulating the prosodic features prior to synthesis. Especially important for convincing prosody are the durations of the synthesised speech sounds. It has been

shown [37, 40] that probabilistic modelling of durations can substantially improve deep generative TTS. This appears especially useful for speech uttered spontaneously in conversation, as considered here, due to its highly diverse and non-deterministic prosodic structure [47]. Inspired by these advances, we introduce a probabilistic duration model coupled with explicit pitch and energy models into the multimodal synthesis architecture. Better duration modelling should help create speech rhythm and timings that allow adequate time for gesture-preparation phases, so that beat-gesture strokes can be distinct and synchronised with the speech. Improved control will not only affect the output speech but also the gestures we generate with it.

### 2.3. Gesture synthesis

Like TTS, deep learning has led to a boom in 3D gesture synthesis from speech text and/or audio [66]. The list of deep generative techniques considered includes GANs [95, 96], normalising flows [4, 5], VAEs [23], VQ-VAEs [102, 103], combinations of adversarial learning and regression losses [20, 26, 53], and combinations of flows and VAEs [86]. Following the impressive performance of text-prompted diffusion models for generating images [77] and human motion [38, 87, 109], diffusion models have seen rapid adoption for 3D gesture-motion generation . As diffusion models require many neural-network evaluations during synthesis, which is slow, flow matching [51] has subsequently been investigated for faster synthesis of high quality output, both for human motion [31, 62] and TTS [25, 48, 63]. Similar to LLMs and large TTS models, recent efforts have also wholly or partly modelled gestures autoregressively as a sequence of discrete tokens [64, 99, 107].

The most recent large-scale comparison of gesture-generation models, the GENEA Challenge 2023 [44], found that the two strongest methods [17, 100] (which are extensions of [7, 98]) were based on diffusion models. Among these, [17] made use of self-supervised text-and speech embeddings from data2vec [8], subsequently aligned with gesture motion using CLIP [72] training, to improve the coherence between gestures and the two speech-input modalities. In addition to modelling beat gestures, the approach recognises the need for additional input modalities to generate representational gestures, such as iconic and deictic pointing [18], for more nuanced and contextually relevant non-verbal communication.

Our data-synthesis pipeline leverages their approach to create synthetic training gestures that well match the synthetic speech text and audio input.

### 2.4. Joint synthesis of speech and gestures

Speech synthesis and gesture generation have traditionally been treated as separate problems, performed on different data by distinct research communities. TTS is mainly developed for read-aloud speech, whereas co-speech gesturing is more closely associated with conversational settings.

Joint synthesis of speech and motion was first considered by [78]. The first neural model was DurIAN [106], which simultaneously generated speech audio and 3D facial expressions, albeit for speech read aloud. [6] trained separate deep-learning TTS and speech-to-gesture systems to synthesise speech and 3D motion for the same speaker and the same (spontaneous) speaking style. This was followed by [94], which investigated adapting and extending AR [79] and NAR [36] neural TTS models to perform joint multimodal synthesis. Their joint models reduced the number of parameters needed over [6], but the best model (the one based on [79]) required complex multi-stage training to speak intelligibly and did not improve quality.

Diff-TTSG [61] advanced joint speech-and-gesture synthesis by employing probabilistic modelling, specifically a strong denoising probabilistic model (DPMs) [82] building on the TTS work in [69]. This model could be trained on speech-and-gesture data from scratch in one go and produced improved results over [94], but internally used separate pipelines for producing the two output modalities, leading to suboptimal coherence between them. Match-TTSG [62] improved on this aspect by using a compact and unified decoder to jointly sample both output modalities. It also used conditional flow matching [51] rather than diffusion, for much faster output synthesis. Experiments found that Match-TTSG improved on the previous best model in all respects, establishing it as the current state of the art.

Most of the above models were trained only on small, parallel multimodal datasets from a single speaker. (The one exception is [94], which required pre-training part of the network on a TTS corpus to produce intelligible output at all.) The results in [62] show that, e.g., the synthetic speech falls short of human-level naturalness, and the quality we find from systems trained on very large datasets. Accordingly, we propose to circumvent the data limitation by using strong unimodal synthesisers to create a large synthetic training corpus for our joint model.

### 2.5. Training on synthetic data

The idea of training deep neural models on the output of other such models has an extensive history. This was originally proposed for classifiers [29], but has subsequently been adapted to generative models, e.g., for TTS [89]. Synthesis (and synthetic data) is also appealing in scenarios where real data is scarce or difficult to obtain, as demonstrated in applications to human poses and motion [90, 108]. It also allows for the creation of diverse and controlled datasets that can enable more accurate and versatile models [35]. We here propose to generalise such approaches by chaining together multiple unimodal synthesisers, to enable training multimodal speech-and-gesture models.

There may be a risk that the individual unimodal synthesisers in the proposed approach could fail to capture mutual information that connects the modalities, since the different synthesisers are likely to be trained on non-overlapping data. This could in turn lead to synthesis artefacts and failure to recreate correlations and dependencies between modalities in systems trained on the final synthetic multimodal corpus. However, recent theoretical and practical results demonstrate that little [55] or no [52, 65] parallel data may suffice for learning joint distributions of multiple random variables (modalities). This suggests that training on corpora generated by synthesisers built from non-overlapping material might not be as risky as it might seem.

## 3. Method

In this section we first describe our method for creating wholly synthetic multimodal datasets for pre-training synthesis models, followed by a description of our modifications to the Match-TTSG architecture to improve durations, prosody control, and multi-speaker data.

### 3.1. Creating synthetic training data

Our pipeline for creating synthetic training data had the following main steps:

1. Generating written sentences in the style of conversational speech transcriptions.
2. Synthesising diverse speech audio from the text.
3. Validating/filtering the synthetic speech audio using automatic speech recognition, and aligning the input text with the synthesised audio.
4. Synthesising gestures from the generated speech audio files and their corresponding time-aligned text.

We provide more detail in the following subsections.

#### 3.1.1 Text generation

The first step was to create text sentences that can form the basis of synthesising multimodal data in a conversational style. For this we utilised GPT-4 [2] and deliberate prompting. Specifically, we prompted the model with a list of 50 text transcriptions sentences from the training split [61] of the Trinity Speech-Gesture Dataset II (TSGD2) [19, 21], each enclosed in triple quotes, followed by a prompt requesting the model to produce 50 additional phrases in the same style (including hesitations and disfluencies as seen in the transcriptions) but ignoring the content. Further prompting then followed, to make the model generate additional output based around different emotions and scenarios, so as to obtain a more diverse material. The emotional categories we provided were: disgust, sadness, fear, frustration, surprise, excitement, happiness, confusion, and denial. Our prompting often gave similar instructions multiple times, since we found that such redundancy led to more realistic

output. The main instruction prompt and a number of example continuations can be found in Appendix A.

We utilised the above procedure to generate a total of 600 phrases, each approximately 250 characters in length. We found that limiting the length of the prompt helps prevent issues with the subsequent speech synthesis, which shows a tendency to produce unintelligible or confabulated output when processing overly long utterances. The 600 generated phrases will be shared in future revisions of the paper.

#### 3.1.2 Speech generation

The next step was to synthesise speech audio from the 600 LLM-generated phrases. For this, we considered multiple TTS systems capable of multi-speaker and spontaneous speech synthesis, including Bark[1], XTTS [15], and ElevenLabs [2]. However, Bark exhibited frequent confabulations and unexpected changes in speaker identity within a single utterance, which seemed problematic for learning to maintain a consistent vocal identity. Although ElevenLabs demonstrated high-quality output, its status as a non-open source and proprietary solution led us to exclude it. Ultimately, we selected XTTS for generating our synthetic speech dataset, due to it combining more consistent synthesis with a research-permissible license. We limited each synthesised utterance to at most 400 XTTS speech tokens, since anything longer than that is virtually certain too long for our prompts, and thus must contain confabulation or gibberish speech. For everything else, default XTTS synthesis hyperparameters were used. In the end, each synthesised audio utterance was around 20–23 seconds long, taking about half that time to synthesise.

In order to obtain more diverse data containing multiple speakers, each of the 600 phrases was synthesised 16 times, once in each of 16 different voices. These voices were selected as a gender-balanced set (8 male and 8 female speakers) from the VCTK corpus [97], and elicited from XTTS by seeding the synthesis of each individual utterance with the audio of longest VCTK utterance spoken by the relevant speaker as an acoustic prompt. These prompting utterances tended to be around 9 seconds long. In total, we thus synthesised $16 \times 600 = 9600$ audio utterances.

Interestingly, despite the spontaneous nature of the input phrases, we found that false starts and fillers explicitly present in the input were sometimes omitted in the XTTS output. This could be partly due to the choice of temperature parameter at synthesis time (the default, 0.65), which favours more consistent and likely output, and partly due to the public English-language training datasets cover read rather than spontaneous speech. Since XTTS furthermore was prompted using a snippet of read-aloud speech audio

---

[1] https://github.com/suno-ai/bark
[2] https://elevenlabs.io/

from VCTK, the output audio tended to sound more like reading than speaking spontaneously.

### 3.1.3 Data filtering and forced alignment

Following speech synthesis, a number of data-processing steps were performed to obtain a suitable dataset for training a strong gesture-generation system. To begin with, all synthesised audio utterances longer than 25 seconds were immediately and permanently discarded, since these overwhelmingly tended to contain issues related to confabulation and the like. The output from XTTS did not have exact fidelity to the text it was prompted with, so automatic speech recognition (ASR) was used to get more accurate input to the gesture-generation system. ASR was performed using Whisper [73], using the `medium.en` model, which has in previous uses proven to be less prone to confabulation than the large variants, whilst providing sufficient accuracy. Interestingly, Whisper tended to prefer British English spelling, possibly since VCTK was recorded in the UK. The ASR derived transcripts then replaced the original TTS input text for each utterance in all subsequent processing.

The gesture-generation system we chose for the final synthesis ([17]) requires word-level timestamps for the text transcriptions. Although we considered several tools that attempt to obtain word timings from Whisper directly, none were sufficiently accurate for our needs. Instead, we obtained the requisite timings using the Montreal Forced Aligner (MFA) [56]. Text input to MFA was processed word-by-word to remove leading and trailing punctuation and to perform case folding to lower case. Utterances that MFA failed to align were also excluded from consideration.

Following the filtering and alignment process, we were left with 8173 audio utterances for our final synthetic dataset, meaning that 1427 utterances (about 15%) were discarded during the filtering step. The remaining data had a total duration of 37.6 hours, which also ended up being the size of the final synthetic training corpus.

### 3.1.4 Gesture generation

We used a recent diffusion-based gesture-generation method [17] that performed well in a large comparative evaluation [44] to generate synthetic gesture data. That system leveraged data2vec [8] embeddings to represent audio input, which help achieve a more speaker-independent representation. On top of that, [44] introduced a Contrastive Speech and Motion Pretraining (CSMP) module, to learn joint embeddings of speech and gesture that can strengthen the semantic coupling between these modalities. By utilising the output of the CSMP module as a conditioning signal within the diffusion-based gesture-synthesis model, the system can generate co-speech gestures that are human-like and semantically aware, thereby improving the quality and appropriateness of the generated gestures to the spoken content. The CSMP module requires word-level timestamps, which is why forced-alignment was performed in Sec. 3.1.3.

Since this paper is focused on multimodal synthesis from data where no interlocutor is present or recorded (i.e., not back-and-forth conversations), interlocutor-related inputs were removed from the architecture. The input is thus an audio track with time-aligned text transcripts. We used the pre-trained weights from [17] for the CSMP module and re-trained the diffusion-based gesture model to comply with the change of input, using the same architecture and learning rate as in the paper. The training was done using two NVIDIA RTX3090 GPUs (194k updates, each with batch size 60) on the subset of the Talking With Hands (TWH) dataset [49] provided in the GENEA 2023 Challenge [44]. We used the trained system to generate text-and-audio-driven gestures for the 8173 previously transcribed synthetic speech utterances, and used Autodesk MotionBuilder after synthesis to retarget the output motion to the skeleton of the TSGD2 data and visualiser in Sec. 4.1. While the synthesised motion encompasses the full body (without fingers), we only consider upper-body motion in this work. Compared to conventional conditioning approaches where audio is represented using mel-spectrograms, the speaker-independent data2vec embeddings in the CSMP module are expected to better handle the differences between natural and synthetic voices during synthesis, thus making it feasible to generate large amounts of gesture data based on synthetic speech without undue degradations due to domain mismatch. This data was used to train the different multimodal synthesis systems considered in our experiments.

## 3.2. Proposed multimodal synthesis system

The current state of the art in joint speech-and-gesture synthesis is Match-TTSG [62], a non-autoregressive model which uses conditional flow matching (OT-CFM) [51] to learn Ordinary Differential Equations (ODEs) with more linear vector fields than continuous-time diffusion models [82] create. Such simpler vector fields offer advantages for easier learning and faster synthesis.

We extend the Match-TTSG framework in three ways:
1. Probabilistic instead of deterministic duration modelling, which can benefit deep generative NAR TTS [37].
2. Additional prosody-prediction modules, which are widely used in NAR TTS [75, 112].
3. A speaker-identity input, as necessary for pre-training on the multispeaker data in the large synthetic training set.

We call the resulting system *MAGI* for *Multimodal Audio and Gesture, Integrated*; see Fig. 2 for a diagram.

For (1), we augment the original Match-TTSG architecture with a probabilistic duration predictor based on OT-CFM, as introduced in [48], to learn distributions over speech and gesture durations. This is trained jointly with

Figure 2. Schematic overview of the proposed MAGI architecture and its prosody predictor.

the rest of the system. It replaces the deterministic duration predictor in Match-TTSG, inherited from [25, 36, 63, 69, 75, 112], and uses the same network architecture.

To learn better prosody correlations and enable control over the output, we drew inspiration from [75, 112] and incorporated two prosody-predictor modules into our system: one for pitch prediction and one for energy prediction, both using the same architecture and hyperparameters as the *variance adaptor* in [75]. Such prosody predictors improve the synthesis as they enable the model to learn a less over-smoothed representation, thereby enhancing the variability of the generated output by conditioning the synthesis process on additional prosodic features [76]. The pitch of the training data utterances was extracted using the PyWorld wrapper for the WORLD vocoder[3] with linear interpolation applied in unvoiced segments to achieve continuous pitch contours for the entire utterances. We employed a bucketing approach similar to [75], separately for pitch and energy, to turn predicted continuous values into embedding vectors to be summed with the text-encoder output vectors. However, in contrast to [75], we performed token-level prediction instead of frame-level prediction for the two prosodic properties, since it has been stated[4] that this improves the synthesis whilst reducing memory consumption.

Like in [69], Match-TTSG includes a projection layer that maps the text-encoder output vectors onto a predicted average output vector per token (sub-phone). These averages are used for the so-called *prior loss* in the monotonic alignment search. The process of sampling the output features (i.e., the flow-matching decoder) is also conditioned on these predicted average vectors. However, the latter can introduce an information bottleneck, since averages do not include information about variance, correlations, or higher moments of the output distribution. To improve information flow we instead condition the MAGI decoder directly on the

last layer of the text-encoder, prior to the projection layer.

Finally, we added a speaker embedding for multispeaker synthesis. Specifically, we used a one-hot speaker vector to represent the 16 different speakers in the synthetic training data. This vector was concatenated to other inputs at multiple stages of the synthesis process, including the text encoder, prosody predictors and decoder. The idea with this was to minimise information loss and ensure coherent output across different speaker identities. Since the concatenated vectors only have 16 elements, the impact on model parameter count is small (an increase of a few thousand).

## 4. Experiments

This section experimentally compares our proposed training method and architecture with the previous state-of-the-art method Match-TTSG [62]. Since this is a synthesis work, the gold standard approach to evaluation – and thus the focus of our experimental validation – is subjective user studies. The experiments closely follows those in previous joint synthesis works [61, 62], which in turn follows established practices in speech [32] and gesture evaluation [44].

### 4.1. Data and systems

To test the effectiveness of our method we carried out 3 different subjective evaluations with systems trained on Trinity Speech-Gesture Dataset II (TSGD2) [22], a dataset containing 6 hours of multimodal data: recordings of time-aligned 44.1 kHz audio coupled with 120 FPS marker-based 3D motion capture, in which a male native speaker of Hiberno-English discusses a variety of topics whilst gesturing freely. The same train-test split of the data was used as in [61], with around 4.5 hours of training data – much less than the 38 hours of synthetic multimodal data we created.

We trained Match-TTSG (**MAT**) containing 30.2M parameters, and MAGI (**MAGI**) containing 31.6M parameters for 300k steps on only the TSGD2 data, we refer to these conditions **MAT-T** and **MAGI-T** respectively. We also took

---

[3] https://pypi.org/project/pyworld/
[4] https://github.com/ming024/FastSpeech2?tab=readme-ov-file#implementation-issues

the same two architectures (albeit with one-hot speaker vectors for Match-TTSG) and first pre-trained them for 200k updates on the synthetic multispeaker data, followed by fine-tuning for 100k updates on TSGD2. We refer to these as **MAT-FT** and **MAGI-FT**. Output samples for held-out sentences were synthesised using 100 neural function evaluations (NFEs; equivalent to number of Euler-forward steps used by the ODE solver) for audio-and-motion synthesis, whilst 10 NFEs were used for the preceding stochastic duration modelling, since it is lower-dimensional and converged more rapidly. Training and synthesis were performed on NVIDIA RTX 3090 GPUs with a batch size of 32.

15 utterances from the held-out set were used to evaluate each modality individually. We used pretrained Universal HiFi-GAN [39] to generate vocoded but otherwise natural speech referred to as **NAT**. We used the same vocoder to generate waveforms from the output mel spectrograms synthesised by the trained multimodal-synthesis systems, while Blender was used to render the motion representations into 3D avatar video, using exactly the same upper-body avatar and visualiser as in [61, 63]. The motion data was represented as rotational representation using exponential maps [24] of 45-dim pose vectors and were downsampled to 86.13 FPS using cubic interpolation to match the frame rate of the mel-spectrograms.

## 4.2. Evaluation setup

To gain an objective insight into the intelligibility of the synthetic speed, we synthesised the test set sentences from TSGD2, which we then passed to Whisper ASR, to use the Word Error Rate (WER) results as an indicator of their intelligibility. For subjective evaluation, user studies are the gold standard when evaluating synthesis methods. Following [61], we used comprehensive evaluation, conducting individual studies of each generated modality. We additionally evaluate the appropriateness of the modalities in terms of each other, to determine how well they fit together.

In our studies, participants had an interface with five unique response choices, with the exact details varying slightly across different investigations. All participants were native English speakers recruited through the Prolific[5] crowdsourcing platform. Each test was designed to last around 20 minutes and participants were compensated 4 GBP (12 GBP/hr) for participation. For the purpose of statistical examination, we converted responses into numerical values. These values were then analysed for statistical significance at the 0.05 threshold using pairwise t-tests.

### 4.2.1 Speech-quality evaluation

To assess perceived naturalness of the synthesized speech, we employed the Mean Opinion Score (MOS) testing ap-

proach, drawing inspiration from the Blizzard Challenge for text-to-speech systems [70]. Participants were asked, "How natural does the synthesized speech sound?", rating their responses on a scale from 1 to 5, where 1 represented "Completely unnatural" and 5 indicated "Completely natural." The intermediary values of 2 to 4 were provided without textual descriptions. Each participant evaluated 15 stimuli per system and 4 attention checks resulting in a total of 525 responses per condition by 35 participants. Fine-tuning with synthetic data led to performance enhancements for both MAGI and MAT, reducing the WER from 13.28% in MAGI-T to 9.29% in MAGI-FT, and from 12.26% in MAT-T to 8.35% in MAT-FT.

### 4.2.2 Motion-quality evaluation

We evaluate motion quality using video stimuli that only visualised motion, without any audio, in order to have an independent assessment of motion quality. This ensures that ratings are not affected by speech and follows the practice of recent evaluations of gesture quality [33, 74]. Similarly to the speech evaluation, participants were asked "How natural and humanlike the gesture motion appear?", and gave responses on a scale of 1 ("Completely unnatural") to 5 ("Completely natural"). The number of stimuli and attention checks were identical to the speech-only evaluation.

### 4.2.3 Speech-and-motion appropriateness evaluation

We finally evaluated how appropriate the generated speech and motion were for each other, whilst controlling for the effect of their individual quality following [33, 45, 62, 74, 105]. For each speech segment and condition, we created two video stimuli: one with the original video and sound, and the other combining the original speech audio with motion from a different video clip, adjusting the motion speed to align with the audio duration. Both videos feature comparable motion quality and characteristics from the same condition, but only one video's motion is synchronised with the audio track, without indicating which video is which.

The test inquired which character's motion most accurately matched the speech in rhythm, intonation, and meaning. Participant ability to identify the correctly synchronised video indicates a strong rhythmic and/or semantic link between generated motion and speech. Following [61] we opted for five response choices instead of the typical three for better resolution. Options were "Left is much better", "Left is slightly better","Both are equal", "Right is slightly better", "Right is much better". For the purposes of analysis, codes in the range of $-2$ to 2 were assigned to each response, as in [61], with $-2$ representing the participant's preference for the mismatched stimulus and 2 the matched stimulus. Participants reviewed motions from 14 of the 15 segments, displayed as 7 screens of pairs of videos, plus

---

[5]https://www.prolific.com/

Table 1. Result of three evaluations showing Mean Opinion Scores (MOS) and 95% confidence intervals.

| Condition | Speech | Gesture | Speech & Gesture |
|---|---|---|---|
| NAT | 4.30±0.06 | 4.10±0.08 | 1.10±0.10 |
| MAT-T | 3.43±0.10 | 3.28±0.11 | 0.52±0.10 |
| MAT-FT | 3.56±0.10 | 3.39±0.09 | 0.56±0.09 |
| MAGI-T | 3.44±0.09 | 3.11±0.10 | 0.51±0.09 |
| MAGI-FT | 3.62±0.08 | 3.52±0.11 | 0.60±0.09 |

two audio and two video attention checks, covering all conditions for these segments. 70 people completed the test, yielding 490 responses per system.

## 5. Results and discussion

Our investigation revealed several key insights into the effect of pre-training and architectural modifications. Pre-training on synthetic data markedly enhanced the quality of synthesised speech, though adjustments to the architecture did not significantly alter its naturalness. Despite this, both MAGI-FT and MAT-FT yielded higher Mean Opinion Scores (MOS), albeit without statistical significance. Notably, the MAGI facilitated greater control over pitch and energy–a feature absent in the original MAT framework. However, despite improvements, the synthesised speech did not achieve the level of naturalness present in the human-recorded speech from the held-out set, see Table 1.

In terms of synthesised gestures, MAGI outperformed other conditions in human-likeness. However, they remained inferior to human-motion reference data. The influence of synthetic data pre-training and the proposed model's architecture on gesture synthesis presented a more nuanced picture. Specifically, pre-training on synthetic data only significantly benefited the proposed model, and, intriguingly, the MAGI enhanced gestures in a larger dataset but had the opposite effect on a smaller dataset. This discrepancy might stem from the prosody predictors in our model being trained on per-phone rather than per-frame data, leading to a scarcity of training data for these predictors in smaller datasets. However, with adequate pre-training on expansive datasets, these models demonstrated better convergence. These findings align with prior speech evaluations, where the novel architecture's advantages were more pronounced following pre-training on a larger dataset.

Further, no model matched the cross-modal appropriateness found in multimodal human recordings, echoing the challenges observed in unimodal gesture synthesis where recent evaluations did not approach the appropriateness of human data [45, 105]. Although MAGI, pre-trained on synthetic data, showcased superior performance, it did not significantly exceed the existing benchmarks in synthesis systems. This observation may be attributed to the inherent difficulty in discerning significant differences in appropriateness, as opposed to naturalness or human-likeness, and the comparison against a robust baseline without alterations that directly influence cross-modal synthesis aspects. Lastly, the accuracy of capturing cross-modal aspects might be least represented in synthetic datasets created from unimodal synthesizers trained on non-cohesive data.

### 5.1. Pitch and energy control

As stated, the proposed multi-stage architecture with separate prosody predictors allows for modifying or substituting the pitch and energy contours before synthesis. This enables direct control of prosodic properties of the speech, with the synthesis process having the option to adjust the gestures to match. On our anonymous webpage cvprhumogen24.github.io/MAGI we provide example videos showing the effect that modifying (scaling) the pitch and energy contours returned by the predictors has on the synthesised output. One can observe that reducing the pitch seems to promote creaky voice, which makes sense from a speech-production perspective and fits earlier findings from autoregressive TTS on spontaneous-speech data [47].

## 6. Conclusion and future work

We have described improvements to the joint and simultaneous multimodal synthesis of speech audio and 3D gesture motion from text. Specifically, we propose pre-training on data synthesised by a chain of strong unimodal synthesis systems to address the shortage of multimodal training data. We also augment the state-of-the-art architecture for speech-and-gesture synthesis, Match-TTSG, with a stochastic duration model, TTS-inspired prosody predictors for controllability, and the ability to perform multi-speaker synthesis. The final model, called Multimodal Audio and Gesture, Integrated (MAGI), is radically smaller than those that generated the synthetic data. Experiments confirm that pre-training on synthetic data significantly improved unimodal speech and gesture quality. The architectural improvements reaped benefits when pre-training on large amounts of synthetic data, with the added prosody control having a clear effect on the audio output.

Relevant future work includes investigating alternative options for mitigating the shortage of multimodal training data, such as pre-training on data lacking one or more of the modalities, incorporating RL-based approaches, particularly effective for generation of situated gestures as in [18], or (following the CSMP methodology [17]) leveraging various self-supervised representations trained on large amounts of data. Possible architectural extensions including flow matching for pitch and energy, and similar control over motion properties such as gesture radius and symmetry [5].

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
