# Fake it to make it: Using synthetic data to remedy the data shortage in joint multimodal speech-and-gesture synthesis

## Supplementary Material

### A. GPT-4 prompts

After supplying 50 example utterance transcriptions from TSGD2, the generation of synthetic phrases was initialised using the following text prompt:

> Take these sentences into account and learn their conversational style ignore the content learn the hesitations and disfluencies (using three dots for long pauses ...) and uh, um and uhm for conversational disfluencies. Generate more sentences like this in form of spontaneous conversational monologues. Remember disfluencies should be either repeating, ..., uh, um or uhm. Make it sound natural as human would converse. Create 50 phrases which mimics how people speak including filled pauses. Make phrases that are highly emotional but realistic.

After that, the model was prompted to generate further phrases corresponding to a variety scenarios and emotions, to obtain more variety. Here are some examples:

- Continue generation for a happy emotion imagine a different scenario of getting your research paper accepted after a difficult and long review process.
- continue generation about confusion emotion talk about getting lost on the way to a new city
- Continue the generation for happy tone and talk about your favourite movie that you recently watched at the theatre and recommend it to people
- Make these sentences a bit smaller just reduce 5–10 words max per sentence. Keep the conversational style and disfluences it is very important to keep those. keep uh, um and pauses, repeats, fixes and other disfluences. continue happy generation talk about a perfect date at your favourite restaurant
- generate more, but start the sentence in a different way. generate some sentences with negation like saying no to things, denying something etc. focus on new topics, you can forget the old topic. talk about people compelling you to learn to play different sports, musical instruments, dance forms and you denying to it.
- continue generation, talk about how excited you are to get a new dream job