# OpenReview forum: "Fake it to make it: Using synthetic data to remedy the data shortage in joint multimodal speech-and-gesture synthesis"
_thecvf.com/CVPR/2024/Workshop/HuMoGen — CVPR 2024 Workshop HuMoGen Submission_

### Official Review · Reviewer_gp7c · 2024-03-30
**An interesting exploration of alleviating Data Shortage in speech-and-gesture synthesis through synthetic data generation. But more convincing experiments for the synthetic data's effectiveness are needed. Tending to a weak acceptance.**

**Rating:** 4
**Confidence:** 5

**Review:**

Data Shortage is a common challenge in speech-and-gesture synthesis. This paper attempts to construct a synthetic dataset and explores its utilization to enhance an existing pre-trained speech-and-gesture model. The clarity and structure of this paper facilitate ease of understanding. The authors employ a subjective evaluation mechanism, which is a standard method in this field.

**Pros**
* Combining multiple large-scale models (GPT-4, XTTS, and Whisper) with a pre-trained speech-and-gesture model to generate synthetic data.
* Adapting concepts from the audio model field to the design of motion models, potentially inspiring future research.

**Cons**
* The evaluation of the synthetic data's effectiveness is not convincing. The model used for synthetic motion generation is trained on the TalkingWithHands (TWH) dataset, implying that the motion distribution is similar to that of the TWH dataset. The MAT-FT/MAGI-FT setup—pre-trained on synthetic data and then fine-tuned using TSGD2—is intuitively equivalent to training on a combination of TWH and TSGD2 datasets. It is straightforward that the performance of this setting is better than MAT-T/MAGI-T that trained on only the TSGD2. But it is hard to prove the effectiveness of the synthetic dataset. More fair and further experiments need to be conducted, such as:
    * Establishing a baseline trained on both TWH and TSGD2 datasets.
    * Utilizing synthetic data to fine-tune the pre-trained model described in Section 3.1.4. If the performance of the model is improved, it indicates that synthetic data works and perhaps enhances the diversity of the original data. This setting is common in NLP.

**Typo**
* Lines 1231-1251: The first letter of each bullet point varies in case.

This paper examines the increasingly important issue of Data Shortage in the data-driven domain of speech-and-gesture synthesis. While the discussion is intriguing, it needs more convincing experiments for support. Consequently, my inclination is towards a weak acceptance.

---

### Official Review · Reviewer_tZYX · 2024-04-01
**Elegantly increasing the amount of labeled data to obtain improved results, but the novelty is limited.**

**Rating:** 4
**Confidence:** 4

**Review:**

This paper uses unimodal synthesis models to create multimodal synthetic parallel training data. Then it trains a joint synthesis model, that outperforms prior art.

Pros:

- Mitigates the shortage in labeled data.
- The authors present a simple and elegant pipeline. Simplicity IS an advantage.
- Experiments demonstrate both the effect of scaling data size and the effect of enhancements in Match-TTSG [62].
- Results outperform prior works, both quantitatively and qualitatively.
- Manuscript is written in a clear and well organized way.

Cons:

- I am afraid the novelty is Incremental: The authors use two existing works to create the synthetic data, with minor changes only. Creating synthetic data is also not a novel idea. The final network that predicts multimodal output is an improved version of Match-TTSG [62], where the improved parts are based on existing ideas..
- Language should be somewhat improved. For example, in several places, the word ‘train’ is more appropriate than the word ‘pretrain’. If this work is accepted, please review the manuscript for any linguistic errors and make necessary corrections.


Altogether, although this work uses existing building blocks, it combines them in an elegant way that induces improved results. Hence, I am inclined for a weak acceptance.

---

### Meta-Review · Area_Chair_ZKhR · 2024-04-05

**Recommendation:** Accept

**Metareview:**

The paper addresses the task of multimodal synthesis of speech and gesture from text.

Pros:
* Well written
* Simple yet effective method
* Quantitative and Qualitative results outperform current work

Cons:
* Limited novelty

**Guidance to authors:** Resolve concerns raised by the authors, particularly the ones related to language.

---

### Decision · Program_Chairs · 2024-04-06

**Decision:**

Accept

**Comment:**

The paper will be published as part of the official CVPR workshop proceedings upon submission of the camera-ready version.